# Tumor Vaccines: Unleashing the Power of the Immune System to Fight Cancer

**DOI:** 10.3390/ph16101384

**Published:** 2023-09-29

**Authors:** Dequan Liu, Xiangyu Che, Xiaoxi Wang, Chuanyu Ma, Guangzhen Wu

**Affiliations:** 1Department of Urology, The First Affiliated Hospital of Dalian Medical University, Dalian 116011, China; liudq@dmu.edu.cn (D.L.); chexiangyu@firsthosp-dmu.com (X.C.); 2Department of Clinical Laboratory Medicine, The First Affiliated Hospital of Dalian Medical University, Dalian 116011, China; wangxiaoxi@firsthosp-dmu.com

**Keywords:** TME, cancer immunology, oncology vaccines, combination therapies, personalized cancer vaccines

## Abstract

This comprehensive review delves into the rapidly evolving arena of cancer vaccines. Initially, we examine the intricate constitution of the tumor microenvironment (TME), a dynamic factor that significantly influences tumor heterogeneity. Current research trends focusing on harnessing the TME for effective tumor vaccine treatments are also discussed. We then provide a detailed overview of the current state of research concerning tumor immunity and the mechanisms of tumor vaccines, describing the complex immunological processes involved. Furthermore, we conduct an exhaustive analysis of the contemporary research landscape of tumor vaccines, with a particular focus on peptide vaccines, DNA/RNA-based vaccines, viral-vector-based vaccines, dendritic-cell-based vaccines, and whole-cell-based vaccines. We analyze and summarize these categories of tumor vaccines, highlighting their individual advantages, limitations, and the factors influencing their effectiveness. In our survey of each category, we summarize commonly used tumor vaccines, aiming to provide readers with a more comprehensive understanding of the current state of tumor vaccine research. We then delve into an innovative strategy combining cancer vaccines with other therapies. By studying the effects of combining tumor vaccines with immune checkpoint inhibitors, radiotherapy, chemotherapy, targeted therapy, and oncolytic virotherapy, we establish that this approach can enhance overall treatment efficacy and offset the limitations of single-treatment approaches, offering patients more effective treatment options. Following this, we undertake a meticulous analysis of the entire process of personalized cancer vaccines, elucidating the intricate process from design, through research and production, to clinical application, thus helping readers gain a thorough understanding of its complexities. In conclusion, our exploration of tumor vaccines in this review aims to highlight their promising potential in cancer treatment. As research in this field continues to evolve, it undeniably holds immense promise for improving cancer patient outcomes.

## 1. Introduction

According to a comprehensive epidemiological survey conducted in 2020, malignancy persistently occupies the top position as the primary instigator of morbidity and mortality globally, with an alarming approximation of 19.3 million novel cancer incidences and nearly 10 million fatalities within that year [1]. The health and medical system confronts formidable challenges. However, a new hope in cancer treatment has emerged in the form of tumor vaccines. Tumor vaccines represent a breakthrough in oncology, leveraging immunotherapy principles to stimulate the body’s immune system against cancer, offering a more targeted approach with fewer side effects compared to conventional treatments like surgery, radiotherapy, and chemotherapy [2].

The evolution of cancer vaccines has unfolded over several decades, with each era marked by pivotal advancements (Figure 1). The journey began in the late 19th century with William Coley, who developed Coley’s toxins after noticing that infections often induced tumor shrinkage in cancer patients [3]. This approach leveraged the immune system to fight cancer—a concept that was further expanded upon by Lewis Thomas and Frank Macfarlane Burnet in the mid-20th century, who introduced the ideas of cancer immunoediting and tumor immunity [4]. The dawn of clinically approved cancer vaccines arrived in 2010 with the FDA approval of sipuleucel-T, the first therapeutic cancer vaccine targeting prostate cancer [5]. The subsequent era, which continues to this day, is characterized by the development of modern cancer vaccines and checkpoint inhibitors, such as a combination therapy approved in 2021 for the treatment of lymphoma [6]. Currently, the development of cancer vaccines emphasizes personalization, with numerous strategies under investigation, including neoantigen vaccines, DNA/RNA vaccines, and viral vectors, aiming to tailor therapies to individual patients’ tumors.

Delving into the mechanistic complexities, the tumor microenvironment (TME), within which tumors exist and evolve, plays a critical role in tumor progression. Consequently, the TME has emerged as a locus of scientific investigation in the arena of anticancer therapeutic drug evolution [7]. Serving as a testament to the innovation of contemporary medicine, tumor vaccines are meticulously contrived to exploit the innate and adaptive capacities of the host immune system to target aberrant cells. These pioneering formulations strategically modulate the TME, thereby judiciously minimizing the collateral devastation inflicted upon healthy tissues [8].

The field of cancer immunotherapy is experiencing significant advancements with the development of diverse tumor vaccines, each with its unique strengths and challenges. Peptide vaccines, though effective in initiating immune responses, often necessitate the use of adjuvants [9]. DNA/RNA-based vaccines allow for the endogenous production of antigens but face hurdles in terms of delivery and expression [10,11]. Viral-vector-based vaccines can stimulate robust immune responses but are constrained by safety concerns and pre-existing immunity [12]. Although dendritic-cell-based vaccines can induce powerful responses, they entail complex production processes and variable patient responses [13]. Whole-cell-based vaccines, while leveraging entire tumor cells to trigger immune responses, grapple with production and standardization issues [14]. An in-depth understanding of these vaccines and their limitations is crucial for their effective deployment in cancer immunotherapy [15]. Current research, cognizant of the limited standalone efficacy of tumor vaccines, is seeking enhanced therapeutic outcomes through their combination with other treatments, such as immune checkpoint inhibitors, chemotherapy, radiotherapy, targeted therapy, and oncolytic virotherapy [16,17,18,19]. Simultaneously, the innovative domain of personalized cancer vaccines is gaining traction, offering a tailored immune response against individual-specific tumors [8]. Despite several challenges, including the complexity of neoantigen identification and resource-intensive production, advances in technology promise a faster and more affordable future for personalized cancer vaccines [20,21,22,23].

This review endeavors to provide a thorough and nuanced exploration of tumor vaccines, linking the foundational mechanisms to their practical clinical applications. As we scrutinize the mechanistic underpinnings of tumor vaccines, our focus is not only to underscore the pivotal role of the TME, but also to dissect the intricate interplay of vaccine mechanisms within this context. Transitioning to the clinical viewpoint, we aim to unravel the character of different tumor vaccines and probe into the state-of-the-art topics that are reshaping the field today. This encompasses the integration of tumor vaccines with other therapeutic methodologies, as well as the surge in pioneering research concerning personalized cancer vaccines. Our aspiration is to cultivate a well-rounded comprehension of tumor vaccines, thereby enriching the ongoing dialogue regarding their potential significance in the realm of cancer treatment.

## 2. Tumor Microenvironment

The tumor microenvironment (TME) is critical in the onset and progression of cancer and is a headache for scientists. It is not only dynamic but also incredibly complex, shaping tumor heterogeneity. The TME is a mixture of multiple cellular and non-cellular components such as cancer cells, stromal cells, immune cells, the extracellular matrix (ECM), and signaling molecules such as growth factors and cytokines [24,25]. The ECM, a tangled web of proteins, glycoproteins, and proteoglycans, lends structure and shapes cell behavior via biochemical and biomechanical signals. When cancer alters the ECM’s composition, it can boost angiogenesis, immune evasion, and therapy resistance [26,27]. For example, increased ECM stiffness, as seen with lysyl oxidase (LOX) overexpression, promotes cellular proliferation and survival through mechanisms like focal adhesion kinase (FAK) activation, while also enhancing tumor invasiveness [28]. Similarly, an accumulation of hyaluronan, often linked to poor prognosis, can bolster tumor growth and facilitate immune evasion [29]. Altered expression of matrix metalloproteinases (MMPs) can remodel the ECM, favoring tumor invasion by releasing growth factors [30,31]. Furthermore, specific ECM proteins like elastin, laminin, tenascin-C, and periostin, when overexpressed, support tumor cell migration and survival [32]. Notably, changes in collagen orientation, resulting in aligned fibers, provide pathways for enhanced tumor cell migration, with such alignment often indicating a higher risk of metastasis [33]. The ECM is also involved in the secretion of various growth factors, like transforming growth factor-β (TGF-β), Interleukin-1β (IL-1β), IL-6, tumor necrosis factor-α (TNF-α), and vascular endothelial growth factor (VEGF), which are secreted by various cells of the TME and can initiate tumor cell growth, survival, migration, angiogenesis, and epithelial–mesenchymal transition (EMT). This is achieved by regulating their specific receptors and stimulating signaling pathways [34] (Figure 2). This intricate interplay between the ECM and tumor cells not only propels cancer progression but also presents challenges in therapy due to factors like drug penetration barriers and the activation of cellular survival pathways.

Similarly, immune cell functions are shaped by a blend of elements. Tumor cells can secrete immunosuppressive cytokines and checkpoint ligands that modify the immune response [35,36], while conditions like hypoxia and acidic pH, resulting from metabolic changes in tumors, can suppress immune activity [37,38]. Concurrently, metabolic competition due to glucose consumption by tumors, alterations in the extracellular matrix, and the recruitment of immunosuppressive cells can inhibit effective immune responses [39]. Direct cell–cell interactions within the TME, the gut microbiome’s influence on tumor immunity, therapeutic interventions, and the presence of chronic inflammation further modulate the balance between pro-tumor and antitumor effects [40,41]. The interplay of these factors determines the complex and dynamic nature of immune responses within the TME.

Regarding immune cells, such as tumor-associated macrophages (TAMs) and T cells, these cells can oscillate in both directions [42,43]. They can either drive tumor growth or stop it. The TME phenotype and cytokines control the expression of all of these cellular and non-cellular components [44]. TAMs, crucial cells in the TME, possess the functional plasticity to either promote or inhibit tumor progression. This is greatly influenced by various cytokines that are present in the TME, leading TAMs to adopt one of two primary phenotypes: the classically activated (M1) macrophages, which have antitumor activity, and the alternatively activated (M2) macrophages which are typically pro-tumor [45]. Certain cytokines, such as interferon-γ (IFN-γ), which is mainly produced by T cells and NK cells, can drive TAMs towards the M1 phenotype. These M1 macrophages are characterized by their ability to present antigens, kill tumor cells, and produce pro-inflammatory cytokines like IL-12 and TNF-α, thereby enhancing antitumor immunity [45]. TNF-α also plays a role in inducing M1 polarization of macrophages, thereby increasing their tumoricidal activity [46]. However, other cytokines, such as IL-4 and IL-13, both of which are mainly produced by Th2 cells, can induce the polarization of macrophages towards the M2 phenotype, which is typically associated with tissue repair, immune regulation, and tumor promotion [47]. Similarly, IL-10, an anti-inflammatory cytokine, can also drive TAMs towards the M2 phenotype, resulting in macrophages that are generally immunosuppressive and produce factors such as VEGF and EGF, which promote tumor growth and angiogenesis [45]. Lastly, TGF-β is known to induce M2 polarization and also has multiple pro-tumor effects, including promoting immune evasion, tissue remodeling, and angiogenesis [48].

T cells, especially cytotoxic T cells (CTLs) and helper T cells, are pivotal in antitumor immunity, with their activation, proliferation, and function being influenced by various cytokines in the TME. Several cytokines propel T cells to combat tumors. Interferon-γ (IFN-γ), primarily produced by Th1 cells and CTLs, enhances the cytotoxic activity of CTLs, promotes their tumor-killing ability, and increases the expression of MHC class I molecules on tumor cells, making them more susceptible to CTL-mediated killing [49]. IL-2 plays a crucial role in T cells’ growth and differentiation, and it is often used in cancer immunotherapy to amplify the immune system’s capacity to fight cancer, mainly through promoting the proliferation and activation of CTLs and NK cells [50]. IL-12 steers the differentiation of naive T cells into Th1 cells, which generate IFN-γ and facilitate CTL-mediated tumor destruction [51]. On the other hand, some cytokines induce tumorigenesis. IL-10, an immunosuppressive cytokine, can hinder the function of effector T cells and antigen-presenting cells, potentially fostering tumor growth by curbing antitumor immunity [52]. TGF-β can inhibit the function of T cells and other immune cells, contributing to tumor immune evasion. It also manifests pro-tumor effects in later cancer stages, promoting tumor cell invasion and metastasis, and initiating angiogenesis [53]. Lastly, IL-6 can aid in tumorigenesis by fostering chronic inflammation, cell survival, and angiogenesis, and it is involved in differentiating T cells into Th17 cells, which in certain contexts are implicated in promoting inflammation and tumor growth [54].

The TME offers promising therapeutic avenues in cancer treatment. Immunosuppressive cytokines like TGF-β, when inhibited, may restore antitumor immune responses and curtail metastasis [55]. Although IL-10 generally suppresses TME immunity, its nuanced roles suggest potential benefits from modulating its levels, while the inhibition of angiogenic and immunosuppressive VEGF has birthed FDA-approved therapies like bevacizumab [56,57]. On the immunostimulatory front, IL-2, known to bolster T-cell growth, has seen therapeutic applications, albeit with side effects, and IL-12’s potent activation of immune cells hints at its combinatorial therapeutic potential [58,59]. Additionally, checkpoint inhibitors targeting the PD-1/PD-L1 axis, such as pembrolizumab, rejuvenate exhausted T cells to counteract tumors [60].

Together, the TME significantly impacts the success of cancer vaccines. While vaccines aim to activate immune cells against tumor antigens, the TME’s immunosuppressive nature can stymie these activated T cells. Factors like altered tumor antigen presentation, reduced MHC molecule expression, and immune checkpoint expression further hinder vaccine-induced responses [61,62]. To enhance cancer vaccine efficacy, researchers are exploring combined therapies with checkpoint inhibitors, methods to reduce the numbers of immunosuppressive cells in the TME, strategies to breach the TME’s physical barriers, the incorporation of potent adjuvants, cytokine modulation, and the development of personalized vaccines tailored to individual tumor antigen profiles [63,64,65]. These multipronged strategies, targeting both the vaccine mechanism and the TME, are steering the direction of next-generation cancer therapies.

## 3. Oncology Vaccines

### 3.1. Cancer Immunology

Adaptive immunity, or acquired immunity, is a highly specific and long-lasting defense against pathogens and abnormal cells, and it is mainly managed by T and B lymphocytes [66]. This type of immunity is famous for its immunological memory, giving lasting protection against things that it has met before, like pathogens or antigens [67]. Studying adaptive immunity’s ins and outs has been vital for creating vaccines and targeted immunotherapies. These breakthroughs offer huge potential for preventing and treating a range of diseases, from infections and autoimmune disorders to cancers [68,69,70].

In cancer’s early stages, the immune system takes part in a process called immunosurveillance, where it hunts down and wipes out abnormal cells, stopping tumors from forming [71,72]. But crafty cancer cells use different tricks to dodge the immune system, helping tumors grow and progress [73]. By studying these immune evasion tactics, scientists have come up with new immunotherapies like immune checkpoint inhibitors (ICIs) and adoptive cell transfer. These innovative treatments aim to boost the body’s ability to fight tumors and overcome cancer’s skill in dodging the immune system [74].

Immunotherapies, like ICIs and chimeric antigen receptor (CAR) T-cell therapy, have totally changed cancer treatment by using the immune system to fight cancer cells [75]. But even with these amazing advances, some patients do not respond well to immunotherapies, or they may even become resistant to them, which means that we still need to learn more about how the immune system and cancer interact [76]. Gaining this knowledge will help us create new immunotherapy strategies and find predictive biomarkers to make patient outcomes even better [77,78].

### 3.2. Mechanism of Action

The fundamental function of cancer vaccines revolves around their capacity to mobilize both innate and adaptive immune responses for the identification, combat, and eradication of neoplastic cells [20]. The following discourse will undertake a comprehensive analysis of their intricate mechanisms of action (Figure 3).

#### 3.2.1. Cellular Immunity

The procedure of eliciting a cellular immune response against cancer, exemplified by the use of cancer vaccines, is complex and sequential. It commences with the delivery of tumor antigens and concludes with the activation of humoral immunity. The following is a detailed breakdown of each phase in this process (Figure 3A).

Introduction of tumor antigens: With cancer vaccines, we are referring to tumor antigens being introduced to antigen-presenting cells (APCs) like dendritic cells. Options include whole tumor cells, peptides, proteins, DNA, mRNA, or even dendritic cells loaded with tumor antigens or packing tumor-derived genetic material. Tumor vaccines play a role in this step. This crucial first step gets the ball rolling in the immune response against cancer [18,79].

Antigen processing and presentation: APCs capture, process, and present tumor-derived peptides on their surface. Dendritic cells efficiently cross-present exogenous antigens to both MHC class I and II molecules, activating both CD8+ cytotoxic T lymphocytes (CTLs) and CD4+ helper T cells [18,80].

Activation of T cells: Presentation of tumor antigens by APCs activates and expands antigen-specific T cells. T-cell activation requires antigen recognition via the T-cell receptor (TCR) and costimulatory signals provided by the interaction between costimulatory molecules on APCs and their receptors on T cells [81,82].

CTLs and helper T cells in action: CTLs directly kill tumor cells by recognizing and binding to MHC class I molecules presenting tumor antigens, while helper T cells produce cytokines that support CTLs’ activation, proliferation, and differentiation. CD4+ helper T cells also provide help to B cells, facilitating antibody production and enhancing the function of APCs and CTLs [2,42].

Stimulation of humoral immunity: Cancer vaccines induce the production of antigen-specific antibodies by B cells. These antibodies target and eliminate tumor cells [15,83].

#### 3.2.2. Humoral Immunity

This procedure initiates with the recognition of tumor antigens and culminates in the implementation of effector functions mediated by antibodies. The following are the details of each stage in this process. (Figure 3B).

Recognition of tumor antigens: Like with cellular immunity, tumor cells have tumor-associated antigens (TAAs) and tumor-specific antigens (TSAs) that the immune system can recognize [84].

Activation of B cells: B cells detect and recognize tumor antigens through their B-cell receptors (BCRs). Add in some costimulatory signals from helper T cells, and B cells are activated to multiply and differentiate [72].

Differentiation of B cells into plasma cells: After activation, B cells morph into plasma cells, specialized for producing vast quantities of antibodies specific to the recognized tumor antigens [84].

Production of antigen-specific antibodies: Plasma cells produce antigen-specific antibodies, which travel through the bloodstream and latch onto tumor antigens on cancer cells [85].

Effector functions of antibodies: Once bound, antibodies wield various strategies against tumor cells, such as the following: Antibody-dependent cell-mediated cytotoxicity (ADCC) (Figure 3A(1–3)): At the first stage, effector cell recruitment: Fc-receptor-bearing immune cells, like natural killer (NK) cells, macrophages, and neutrophils, are drawn to the tumor site via interactions between their Fc receptors and the Fc segment of the antibody bound to the malignant cell [86,87]. At the next stage, formation of immunological synapses, degranulation, and release of cytotoxic molecules: Immunological synapses are formed when Fc receptors on effector cells interact with the Fc portion of cancer-bound antibodies, enabling the targeted release of cytotoxic molecules [88]. Effector cell degranulation ensues, culminating in cancer cell elimination via perforin and granzymes [89]. Finally, phagocytosis: Post-apoptosis, macrophages engulf the cancerous debris, facilitating clearance and preventing the dissemination of harmful cellular components [90].Complement-dependent cytotoxicity (CDC) (Figure 3B(1–4)): At the first stage, complement activation and cascade: Bound antibodies’ Fc portions engage C1q (complement component 1, q subcomponent), instigating the classical complement pathway and the formation of the C3 convertase enzyme complex [91,92]. At the next stage, membrane attack complex (MAC) assembly: Complement protein cleavage and activation yield the MAC, comprising C5b, C6, C7, C8, and C9 [93]. After that, cell lysis: MACs perforate cancer cell membranes, causing uncontrolled ion and water movement, cellular lysis, and death [93]. Finally, opsonization and phagocytosis: Cancer cells are targeted for destruction by phagocytes when complement activation promotes opsonization [92].Neutralization of growth factors and inhibition of signaling pathways: Antibodies obstruct tumor cell proliferation-promoting growth factors and impede signaling pathways that are crucial for cancer cell survival and invasion [15].

## 4. Types and Characteristics of Cancer Vaccines

### 4.1. Peptide Vaccines 

Peptide vaccines represent a potential cancer immunotherapy approach that employs short amino acid sequences originating from tumor-specific or tumor-associated antigens (TAAs) to evoke targeted immune reactions against malignant cells [94]. TAAs, including differentiation antigens, overexpressed antigens, cancer/testis antigens, and mutated antigens, present feasible targets for multiple immunotherapeutic techniques, such as cancer vaccines, adoptive T-cell therapies, and immune checkpoint inhibitors [95,96]. Peptide vaccines offer several benefits, including ease of synthesis, specificity, and a favorable safety profile due to their reduced likelihood of triggering autoimmune responses compared to whole-protein or live attenuated vaccines [97,98]. Nevertheless, they encounter limitations such as suboptimal immunogenicity, rapid in vivo degradation, weak CD4+ T-cell responses, and challenges pertaining to immune evasion and tumor-induced immune suppression [97]. To tackle these limitations, strategies encompass the incorporation of adjuvants, peptide sequence optimization, utilization of carriers to enhance stability and immunogenicity, and the combination of peptide vaccines with other immunotherapies [94].

At present, the relatively mature peptide vaccines include Nelipepimut-S (NeuVax), CIMAvax-EGF, and MUC1-based peptide vaccines. Nelipepimut-S, also known as NeuVax, is a peptide vaccine targeting HER2/neu-expressing cancer cells, primarily focusing on early-stage HER2 1+ and 2+ breast cancer patients who are ineligible for standard HER2 therapies [99]. It combines the E75 peptide from HER2/neu with GM-CSF as an adjuvant for a heightened immune response [100]. Although its scope may cover other HER2/neu cancers like ovarian and gastric cancers, its clinical development hit snags [101]. Recent phase III trial results show that Nelipepimut-S still exhibits good efficacy and tolerability in breast cancer patients [102]. Similarly, CIMAvax-EGF, targeting the epidermal growth factor for non-small-cell lung cancer (NSCLC), merges recombinant EGF with a protein carrier. It has shown promise in prolonging the lives of late-stage lung cancer patients, with phase III trials completed [103,104]. Additionally, MUC1-based peptide vaccines focus on the aberrantly expressed glycoprotein in cancers like breast and pancreatic cancers [105]. Though some have reached phase I and II trials, with promising safety and immune indicators, eliciting robust clinical responses remains complex, leading to the exploration of combination therapies [106,107]. Collectively, these vaccines represent cutting-edge cancer treatment approaches, each with its unique targets and development stages.

### 4.2. DNA/RNA-Based Vaccines

DNA/RNA-based tumor vaccines operate on the principle of delivering genetic material that encodes tumor antigens to host cells. This approach, leveraging various vectors such as viral vectors, lipid nanoparticles, or naked nucleic acids, stimulates the immune system to recognize and destroy cancer cells, leading to an adaptive immune response [108]. Compared to traditional methods, these vaccines confer several advantages, including safety, ease of manufacture, robust immune response induction, and amenability to modifications. They also afford the possibility of personalization to cater to each patient’s unique tumor profile [109,110]. Despite their promise, challenges such as the efficient delivery and uptake of DNA/RNA into cells, the risk of autoimmune responses, and the constraints of cost and time for personalization pose significant hurdles [111,112,113].

However, recent advances, including several vaccines in clinical trials and technological progress enhancing the efficacy of vaccine delivery, are encouraging [114]. The field looks forward to leveraging advancements in genomic sequencing, bioinformatics, and nanotechnology to surmount current limitations, aspiring to blend these potent vaccines with other immunotherapeutic strategies for comprehensive cancer eradication [115,116]. Among them, techniques such as lipid nanoparticles (LNPs), which have been seminal for mRNA COVID-19 vaccines, are being adapted for cancer vaccine development to boost the delivery of tumor-specific antigens [117,118]. Electroporation and viral vectors (like adenoviruses) enhance the uptake of DNA/RNA, while non-viral nanocarriers and microneedle patches aim to augment this delivery without inducing strong anti-vector responses [119,120]. To mitigate autoimmune risks, researchers emphasize tumor-specific antigen selection, sequence optimization to reduce cross-reactivity, and transient expression techniques, such as those inherent to mRNA vaccines [121,122]. Furthermore, tolerance-breaking adjuvants and nanoparticles tailored for targeted delivery are being harnessed to fine-tune the immune response, maximizing antitumor efficacy while minimizing collateral damage to healthy tissues [121,123]. Collectively, these innovations underscore the evolving landscape of cancer vaccine design, balancing potent tumor targeting with patient safety. CV9104 is an mRNA-based cancer vaccine targeting prostate cancer [124]. Its development reached a phase II trial for metastatic castration-resistant prostate cancer (mCRPC) [124]. This vaccine represents the innovative utilization of mRNA in oncology.

### 4.3. Viral-Vector-Based Vaccines

Viral-vector-based tumor vaccines are a promising development in cancer immunotherapy [11]. These vaccines leverage the innate ability of viruses to infiltrate host cells and efficiently deliver tumor antigens, sparking an intense, targeted immune response [125,126,127]. This approach prompts host cells to produce tumor-specific or associated antigens after infection, resulting in these antigens being displayed to T cells and subsequently initiating a robust defense against tumor cells [128]. One of the significant advantages of these vaccines is their capacity to trigger potent cellular and humoral immune responses. They can also be engineered to express numerous tumor antigens, extending their range and enhancing the potency of the immune response [129,130]. Despite these benefits, certain challenges need to be addressed, such as the impact of pre-existing immunity to the viral vector and the logistical issues associated with large-scale production [131,132].

Among these challenges, pre-existing immunity to viral vectors poses a significant challenge for their use in tumor vaccines, as the immune system might neutralize the vector before its therapeutic action [133]. To counteract this, researchers are exploring a range of strategies: using rare or novel viral vectors with limited human exposure, pseudotyping to change viral envelope proteins, employing a heterologous prime–boost strategy with different vectors, making genetic modifications to the viral capsid to reduce recognizability, co-administering with immune modulators to transiently suppress certain immune responses, opting for non-intravenous delivery routes like intratumoral administration to avoid high antibody concentrations, and adjusting dosages, either by using a high vector dose to overcome neutralization or by administering repeated low doses to evade immune detection [134,135,136,137]. Additionally, adjuvants are being explored to shift the focus of the immune response from the vector to the delivered tumor antigen [138,139]. These multifaceted approaches aim to optimize the efficacy of viral-vector-based tumor vaccines in the face of pre-existing immunity.

OncoVEXGM-CSF, or T-VEC, is an oncolytic HSV-1 vaccine modified for tumor selectivity and GM-CSF production, primarily targeting melanoma [140]. It gained FDA approval for unresectable recurrent melanoma following a successful phase III trial [140]. CG0070, another adenovirus-based vaccine, was engineered for selective replication in Rb-pathway-defective cancer cells and targets bladder cancer [141]. LV305, a lentivirus-based vaccine, delivers the NY-ESO-1 antigen gene to dendritic cells, aiming at NY-ESO-1-expressing cancers like melanoma and sarcoma [142]. JX-594, or Pexa-Vec, a vaccinia-virus-based vaccine, is modified to express GM-CSF and selectively target cancer cells with high thymidine kinase activity, and it has undergone several trials, including a phase III trial for hepatocellular carcinoma [143,144,145]. These represent innovative intersections of virotherapy and immunotherapy in oncology.

### 4.4. Dendritic-Cell-Based Vaccines

Dendritic cells (DCs), instrumental in mediating the immune response by linking the innate and adaptive immune systems, and vital in antigen presentation and subsequent T-cell activation, have found substantial relevance in cancer immunotherapy strategies [18,146]. This stems from DC-based cancer vaccines, which employ DCs loaded with tumor-associated antigens (TAAs) to prompt a robust immune response against cancer cells [147]. The approaches for loading these DCs with TAAs are multifaceted, ranging from the use of tumor lysates and synthetic peptides to mRNA encoding tumor antigens [148]. Clinical trials have highlighted the promise of these DC-based vaccines in a variety of cancers. Despite this exciting potential, challenges persist, including technical difficulties in DC vaccine production, vaccine potency variations, the immunosuppressive tumor microenvironment, and the lack of reliable biomarkers for patient selection [149]. However, recent advancements in personalized cancer immunotherapy, such as neoantigen-based vaccines, present promising opportunities for DC-based vaccines, and combining these with other treatments may enhance their efficacy [8]. In a recent study, researchers introduced a metabolic glycan labeling technique using azido sugars for the enhancement of DC vaccines [150]. This method not only boosts DC activation and antigen presentation but also facilitates the efficient conjugation of cytokines [150]. Furthermore, it holds promise for broad applications across various tumors, provides a platform for modulating interactions between DCs and other immune cells, and amplifies the antitumor efficacy of dendritic cell vaccines.

Outstanding representatives include Provenge and DCVax-L. Provenge (sipuleucel-T) is an FDA-approved autologous cellular immunotherapy for advanced prostate cancer [5]. It uses a patient’s peripheral blood mononuclear cells (PBMCs) exposed to a fusion protein, PA2024, which combines an antigen from prostate cancer cells with an immune activator, GM-CSF, priming an immune response against prostate cancer cells expressing the antigen [151,152,153,154,155]. On the other hand, DCVax-L is an autologous dendritic cell vaccine for glioblastoma multiforme (GBM) [156]. The vaccine is prepared by loading a patient’s dendritic cells with tumor lysate from their own tumor tissue, enabling the immune system to recognize and attack corresponding cancer cells [156]. Both vaccines harness dendritic cells to target cancer, but their clinical journeys and disease targets differ (Table 1).

### 4.5. Whole-Cell-Based Vaccines

Whole-cell-based vaccines offer a comprehensive approach to cancer immunotherapy by incorporating a vast array of tumor-associated antigens to stimulate a potent immune response. Mechanistically, these vaccines utilize irradiated tumor cells (autologous or allogeneic) to expose the immune system to the full antigenic repertoire of the tumor [157], leading to the induction of specific and polyvalent immune responses against a range of tumor antigens [158]. This strategy presents a broad spectrum of known and unknown tumor antigens, avoiding antigen loss or downregulation—a typical escape mechanism employed by tumors [157]—and bypassing the need to identify specific antigens for each patient, which can be time-consuming and costly [159]. However, there are limitations. The production of autologous whole-cell vaccines can be labor-intensive and personalized, necessitating the isolation and culture of tumor cells from each patient [159]. These vaccines often require co-administration with adjuvants or immunomodulatory agents to boost their immunogenicity, given that the immunosuppressive tumor microenvironment can limit vaccine efficacy [8,159]. Concerns also linger about potential autoimmunity induced by self-antigens in the vaccine formulation [158]. Thus, while whole-cell-based vaccines offer a promising approach to cancer immunotherapy, further optimization and refinement of these strategies are required to address these challenges and limitations.

Representatives include GVAX, Canvaxin, and Oncophage. GVAX is a whole-cell tumor vaccine, utilizing tumor cells genetically modified to secrete GM-CSF (an immune stimulant), and has been explored for cancers like pancreatic and prostate cancers, with mixed outcomes in later-phase trials [160,161]. Canvaxin, aimed at melanoma, combines irradiated autologous and allogeneic melanoma cells with the BCG adjuvant, but it failed to show significant survival benefits in a phase III trial for advanced melanoma [115,162]. Oncophage (Vitespen) is derived from patient-specific tumor heat shock proteins (HSPs) and primarily targets renal-cell carcinoma and melanoma [163,164]. It completed phase III trials with mixed results but secured approval in Russia for the treatment of kidney cancer [165]. While these vaccines showcase varied cancer immunotherapy strategies, each has faced challenges in late-stage clinical evaluations. (Table 1)

**Table 1 pharmaceuticals-16-01384-t001:** Below is a tabular list of various tumor vaccines in the last decade.

Types of Tumor Vaccines	Strengths	Weaknesses	Examples	Mechanisms of Action	Effects	Limitations	References
Peptide vaccines	Specific to tumor antigens Low toxicity Easily synthesized and scalable	Limited to known antigens May not induce a robust immune response alone	Nelipepimut-S (NeuVax)	HER2-derived peptide vaccine	Activation of T-cell response	Limited overall survival improvement	[166]
			CIMAvax-EGF	EGF-based peptide vaccine	Inhibition of EGF signaling	No direct tumor targeting	[103]
			MUC1-based peptide vaccine	Targeting MUC1 tumor-associated antigens	Enhanced immune response	Heterogeneous patient response	[167]
DNA/RNA-based vaccines	Can encode multiple antigens Flexibility in design Stable and easy to produce	Delivery into cells can be challenging Risk of integration (for DNA) May induce autoimmune responses	CV9104 (CureVac)	Uses mRNA to encode six antigens overexpressed in prostate cancer	Induced antigen-specific immune responses in early clinical trials	Efficacy in late-stage trials yet to be established; possibility of inducing autoimmune responses	[168]
Viral-vector-based vaccines	Efficient cell entry and expression Can induce strong immune responses	Pre-existing immunity to the viral vector can reduce effectiveness Potential for off-target effects	Adenovirus-based vaccines (OncoVEXGM-CSF, CG0070)	Adenoviruses are modified to express a tumor-specific antigen or an immunomodulatory molecule; these stimulate an immune response against the tumor	Effective in stimulating an immune response against the tumor	Immune response to the viral vector can limit repeat dosing	[169]
			Lentivirus-based vaccines (LV305)	Lentiviruses are engineered to deliver tumor-specific antigens to dendritic cells to stimulate a T-cell response	Successful in initiating T-cell responses	Safety concerns over integration into the host genome	[23]
			Vaccinia-virus-based vaccines (JX-594)	Vaccinia viruses are genetically engineered to express a tumor antigen and/or immunostimulatory molecule; they can directly lyse cancer cells	Showed antitumor activity and were well tolerated in clinical trials	Immune response to the viral vector can limit its effectiveness	[170]
Dendritic-cell-based vaccines	Tailored to individual patients Induce potent T-cell responses	Labor-intensive and costly production Requires patient-specific infrastructure	Provenge (Sipuleucel-T)	The patient’s own dendritic cells are exposed to a fusion protein (prostatic acid phosphatase linked to an immune cell stimulating factor)	Extended overall survival in metastatic castration-resistant prostate cancer	Limited clinical benefits, high cost, and complex manufacturing process	[5]
			DCVax-L	Autologous dendritic cells are pulsed with tumor lysate	Prolonged progression-free survival in glioblastoma multiforme (GBM) patients	Not FDA-approved; requires personalized manufacturing	[171]
Whole-cell-based vaccines	Broad range of tumor antigens presented Mimics natural infection	Complex manufacturing Potential for tumor cell growth if not fully inactivated	GVAX	Utilizes autologous/allogeneic tumor cells that have been genetically modified to secrete the immune-stimulating cytokine GM-CSF	Demonstrated a significant immune response against cancer, studied in various types of cancer, including pancreatic and prostate cancers	Production can be labor-intensive and personalized; often requires co-administration with adjuvants or other immunomodulatory agents to enhance their immunogenicity	[159,172]
			Canvaxin	Allogeneic melanoma cells mixed with Bacillus Calmette–Guérin (BCG) to stimulate immune response	Intended for melanoma treatment, but development discontinued due to insufficient effectiveness	Limited efficacy; potential for BCG-related side effects	[173]
			Oncophage (Vitespen)	Uses heat shock proteins (gp96) derived from the patient’s tumor as an autologous vaccine	Showed efficacy in extending disease-free survival in certain patients with kidney cancer and melanoma	Not universally effective; personalized manufacturing can be labor-intensive	[174,175]

The preceding table encapsulates seminal instances of assorted classifications of cancer vaccines, encompassing peptide-based, DNA/RNA-based, viral-vector-based, dendritic-cell-based, and whole-cell-based vaccines. Each paradigm is delineated in exhaustive detail, supplemented by pertinent bibliographical citations for subsequent scholarly inquiry.

### 4.6. Another Cancer Vaccine Therapy: In Situ Cancer Vaccines

In situ cancer vaccines represent a therapeutic approach where the tumor inside a patient’s body is directly targeted to serve as its own vaccine [63]. Rather than extracting tumor cells for external processing and reintroduction, in situ vaccines stimulate the immune system by damaging the tumor in its native environment [63]. As the tumor cells die, they release antigens, which are then recognized by the immune system. Often, this is achieved by injecting immune-stimulating agents or oncolytic viruses into the tumor [176,177]. This not only aims to destroy the immediate tumor but also primes the immune system to recognize and combat tumor cells elsewhere in the body [176]. In situ cancer vaccines have shown promise in preliminary studies.

### 4.7. Influencing Factors of Tumor Vaccines

Boosting the power of cancer vaccines is a top priority for researchers, who are diving deep into adjuvants and combination therapies to ramp up immune responses, outsmart tumor immune evasion, and prevent cancers from coming back [178,179]. The efficacy of cancer vaccines hinges on several factors, including picking the right antigens, choosing the adjuvants wisely, and using the best delivery systems [2]. Antigen selection is of paramount significance. The antigens should be specific to the tumor, or associated with it, so that the immune response zeroes in on cancer cells without harming healthy tissues [180]. Plus, the chosen antigens need to be highly immunogenic and able to stimulate both CD8+ cytotoxic T cells and CD4+ helper T cells for a strong, long-lasting attack against tumors [15]. Adjuvants help by making cancer vaccines more immunogenic. They stimulate the innate immune system, encourage antigen uptake by APCs, and help activate and expand antigen-specific T cells [181,182]. There are different types of adjuvants, like alum, toll-like receptor (TLR) agonists, and cytokines, each with unique mechanisms of action and varying effectiveness [183,184]. The latest research shows that TLR agonists, such as TLR9 and TLR7/8 agonists, have shown promise by bolstering antigen-presenting cells and intensifying immune responses [185]. Researchers have developed a nanosystem that can inhibit a process called MerTK-mediated efferocytosis. This inhibition leads to the release of immunogenic contents into the tumor microenvironment, potentially boosting the body’s natural defenses against the tumor [186]. Similarly, STING agonists enhance dendritic cell activity, boosting T-cell responses against tumors [187,188]. Oncolytic viruses, while serving as direct antitumor agents, also act as adjuvants by releasing tumor antigens within an inflammatory milieu [132]. These recent breakthroughs encapsulate the dynamic progression in adjuvant research, aiming to optimize the immune system’s potency against tumors.

## 5. Combination Therapies

Combining cancer vaccines with other therapies has emerged as a promising strategy to enhance the overall therapeutic efficacy and overcome the limitations of single-agent treatments [189]. Cancer vaccines, which aim to stimulate a patient’s immune system to recognize and attack tumor cells, may benefit from being combined with other immunotherapies, such as immune checkpoint inhibitors, to boost immune responses and counteract immunosuppressive mechanisms within the tumor microenvironment (TME) [190].

Several types of combination therapies involving cancer vaccines and other treatment modalities have been explored in recent years, such as combining cancer vaccines with chemotherapy, targeted therapies, and radiation therapy [189]. These combination approaches hold significant promise for optimizing cancer treatment outcomes and providing more effective, personalized therapy options for patients [74].

### 5.1. Cancer Vaccine + Immune Checkpoint Inhibitors

These vaccines can be combined with other treatments to enhance their effectiveness. Immune checkpoint inhibitors like pembrolizumab (Keytruda), nivolumab (Opdivo), and ipilimumab (Yervoy) disable immune checkpoints, thereby unleashing a more potent attack on cancer cells [191,192,193,194]. This combination hopes to enhance recognition of cancer cells (via the vaccine) and amplify the immune response (via the checkpoint inhibitors) [5,178].

This combined approach has shown promise in preclinical models and early clinical trials by generating tumor-specific T cells and preventing their exhaustion [195]. The mechanism behind these effects is that cancer vaccines aim to boost T cells’ recognition of tumor antigens, but this immune response can be dampened by the tumor’s evasion mechanisms [196]. Enter immune checkpoint inhibitors, which block inhibitory checkpoints (PD-1 and CTLA-4) on T cells, essentially “releasing the brakes” and amplifying their antitumor activity [197]. By combining cancer vaccines, which enhance the number of tumor-recognizing T cells, with checkpoint inhibitors that ensure that these T cells are not suppressed, there is a synergistic boost in the antitumor immune response. Preliminary studies suggest that this combination augments tumor attack, potentially leading to improved patient outcomes [198,199,200,201].

### 5.2. Cancer Vaccine + Chemotherapy

Chemotherapy is a destructive force against cancer cells, hindering their growth and division, but may also inadvertently harm rapidly dividing normal cells such as those in bone marrow, the digestive tract, and skin [202]. The potential synergy between cancer vaccines and chemotherapy arises from some chemotherapeutic agents inducing immunogenic cell death, increasing the visibility of dying cancer cells to the immune system and potentially enhancing the efficacy of cancer vaccines [203]. Several chemotherapeutic agents have been identified to potentially enhance the efficacy of cancer vaccines due to their immunomodulatory effects. For instance, cyclophosphamide and temozolomide can deplete immune-suppressing regulatory T cells (Tregs), creating a more receptive tumor environment for vaccine action [204]. Docetaxel, used for cancers like breast and prostate cancers, can bolster antigen presentation, thereby enhancing immune recognition of tumor cells [205]. Gemcitabine targets and reduces myeloid-derived suppressor cells (MDSCs) [206]. When combined with cancer vaccines, these agents can modify the tumor environment, diminish immune suppression, or amplify the immune response against tumors, although the choice of combination depends on multiple factors, including cancer type and patient health [203].

However, there are substantial challenges to this approach. Determining the optimal timing and dosage of chemotherapy in relation to cancer vaccines remains a complex task [159]. The side effects of both chemotherapy and cancer vaccines, including chemotherapy’s often severe systemic side effects such as fatigue, infection, hair loss, and nausea, are a significant concern [207]. Furthermore, the treatment’s responsiveness is limited, as not all cancer types respond well to chemotherapy or cancer vaccines, with variability in individual patient responses adding to the complexity of treatment plans [208,209]. Additionally, the complexity of the tumor microenvironment, which can evolve various mechanisms to resist or evade treatment, may limit the effectiveness of these combined therapies [210].

### 5.3. Cancer Vaccine + Radiotherapy

Radiotherapy employs high-energy particles or waves, such as X-rays, gamma rays, electron beams, or protons, to annihilate or damage cancer cells. This radiation induces small breaks in the DNA inside cells, inhibiting their growth and division, and eventually leading to their death [211]. When combined with tumor vaccines, these treatments might produce a synergistic effect, with radiotherapy potentially leading to the release of cancer cell antigens and stimulating the immune system, thereby enhancing the effectiveness of cancer vaccines [212,213,214]. Research indicates that radiotherapy exerts both cytotoxic and immunomodulatory effects on the tumor microenvironment. Beyond directly damaging tumor cells, RT induces immunogenic cell death, leading to the release of damage-associated molecular patterns (DAMPs) [215]. These DAMPs serve as “danger signals”, enhancing dendritic cell function and fostering antitumor immune responses. Concurrently, radiotherapy damages the tumor vasculature, increasing its permeability due to direct effects on endothelial cells and the upregulated release of VEGF from irradiated tumor cells [215,216]. This can lead to both transient improvements in oxygen and nutrient delivery and enhanced immune cell infiltration into the tumor.

However, this combination approach has its limitations. Not all patients or cancer types respond well to either radiotherapy or cancer vaccines, making the efficacy of this approach unclear in a broad population [217]. The optimal timing and dosage of radiotherapy relative to cancer vaccines are not well understood, posing a risk of radiotherapy killing immune cells stimulated by the vaccine, and thereby reducing the effectiveness of the treatment [218]. Both treatments can cause side effects, such as skin changes, fatigue, and other symptoms for radiotherapy, and usually mild but possibly flu-like symptoms for cancer vaccines [63], and some tumors may develop resistance to radiotherapy, which could limit the effectiveness of this combined approach [219].

### 5.4. Cancer Vaccine + Targeted Therapy

Compared to cancer vaccines, targeted therapies obstruct specific proteins or processes that aid in cancer growth and progression, offering a more cancer-cell-selective approach compared to traditional chemotherapy and resulting in fewer side effects. Notable targeted therapies include small-molecule inhibitors, like Gleevec (imatinib), and monoclonal antibodies, like Herceptin (trastuzumab) [220,221,222]. When utilized in combination, targeted therapies aim to inhibit cancer cells’ proliferation and survival, rendering the cancer cells more susceptible to the immune response provoked by the cancer vaccine. Studies have revealed the potential of this combination, with targeted therapies able to modulate the tumor microenvironment, thereby possibly enhancing the effectiveness of the vaccine-stimulated immune response [223] and helping to prevent or delay resistance to targeted therapies [2]. However, challenges and limitations remain, including the development of resistance to targeted therapies over time [224], potential side effects ranging from mild skin rashes or diarrhea to severe liver toxicity or heart problems [225], limited responsiveness in certain cancer types or patients [208], and the complex and not fully understood interaction effects between cancer vaccines and targeted therapies, which could potentially interfere with the vaccine-stimulated immune response [16].

### 5.5. Cancer Vaccine + Oncolytic Virotherapy

Oncolytic virotherapy constitutes a novel paradigm in the therapeutic approach towards malignant neoplasms, exhibiting a mechanism of action that distinguishes it from traditional tumor vaccines. It capitalizes on the unique capabilities of selected or genetically engineered viruses, which are orchestrated to specifically target and eradicate neoplastic cells [19,226]. Upon administration, these oncolytic viruses infiltrate the patient’s system, subjugating cancerous cells and commandeering their biological machinery for viral replication, consequently leading to cell lysis [227,228]. This lysogenic cycle not only facilitates direct oncolysis but also liberates tumor-specific antigens, providing a catalyst for the patient’s immune system to mount an anticancer response—an underpinning that is shared with the concept of tumor vaccines [132]. This dual-action mechanism that harmonizes direct cellular destruction with immune activation embodies a promising pathway in the realm of cancer therapy. The dual-action mechanism encompasses direct tumor cell lysis, releasing tumor-associated antigens, and the unveiling of damage-associated molecular patterns (DAMPs) and pathogen-associated molecular patterns (PAMPs). These elements activate both the innate and adaptive arms of the immune system, enhancing antitumor responses. As newly assembled viral entities continue their onslaught against other malignant cells, a self-propagating cycle is established.

Recent advancements in this burgeoning field have entailed the exploration of diverse oncolytic virus models, such as the *Parapoxvirus ovis* model, known to induce an immunogenic form of cell death termed pyroptosis [229]. Scientific investigations have also scrutinized the immunostimulatory effects of these bioengineered viruses and combinational therapeutic strategies that kindle pyroptosis, consequently fostering potent antitumor activity [230,231,232]. These engineered viruses have some potential in the delivery of antitumor drugs [233]. Pioneering therapeutic strategies, such as the KISIMA/VSV-GP heterologous prime–boost methodology and the development of adenovirus-based tumor vaccines, have further emphasized the potential of oncolytic virotherapy as a formidable armament in the arsenal of cancer immunotherapy [234].

Cancer vaccines and oncolytic virotherapy offer a potential synergistic approach for cancer treatment. Cancer vaccines introduce cancer-specific antigens into the body, training the immune system to recognize and attack cells displaying these antigens [5]. Concurrently, oncolytic virotherapy uses engineered viruses that selectively infect and eliminate cancer cells, subsequently releasing tumor antigens and new viral particles that can infect nearby cancer cells, thereby stimulating an immune response [19]. In combination, the cancer vaccine’s potential enhancement of the immune response, coupled with the direct cellular damage from the oncolytic virus, could increase therapeutic effectiveness. Initial studies suggest that this combination can result in a more robust and long-lasting immune response against tumors, even potentially overcoming some immune evasion tactics employed by cancer cells [235]. Nevertheless, this approach is not without limitations. The immune system could respond to the oncolytic virus, reducing its cancer-killing effectiveness [236]. Additionally, delivering both oncolytic viruses and cancer vaccines to the tumor site, particularly in solid tumors, is challenging [237]. The diverse nature of cancers and variability in patient responses can limit the overall responsiveness of this combined therapy [238]. Finally, safety is a significant concern, as both oncolytic virotherapy and cancer vaccines can cause side effects, with the former potentially leading to severe or life-threatening reactions in rare instances [239]. Recent advancements will change this domain. Viruses are now engineered for heightened tumor specificity, some are armed with therapeutic genes to turn tumors into producers of anticancer agents, and combinations with treatments like immune checkpoint inhibitors are showing synergistic effects [240,241]. Moreover, refined genetic engineering techniques have improved the safety profiles of these oncolytic viruses, making them more amenable for therapeutic applications [242,243]. This means that they have reduced virulence in non-target tissues and minimized side effects.

## 6. Personalized Cancer Vaccines

Personalized cancer vaccine therapy is an innovative approach that tailors cancer treatment to a patient’s unique tumor profile. The process can be outlined as follows (Figure 4).

### 6.1. Tumor Sample Collection

A critical step in this process is the collection of tumor samples, which provide the essential genetic material needed to tailor the vaccine to the individual’s specific cancer [244]. The latest research advances have highlighted the importance of obtaining high-quality, well-preserved samples through minimally invasive techniques such as fine-needle aspiration or core-needle biopsy [245]. Additionally, there is a growing emphasis on collecting tumor samples at multiple timepoints throughout the course of treatment to account for the inherent heterogeneity of tumors and the potential for evolving cancer mutations [246]. The integration of these cutting-edge technologies and best practices in tumor sample collection is essential for maximizing the success of personalized cancer vaccine therapy [20].

### 6.2. Sequencing and Analysis

Sequencing and analysis empower researchers and clinicians to identify tumor-specific mutations and neoantigens that may function as potential therapeutic targets [20]. The development of next-generation sequencing (NGS) technologies, such as whole-exome and whole-genome sequencing, has greatly sped up the process and increased the precision of pinpointing tumor-specific mutations [177,247]. Concurrently, innovative computational approaches like machine learning algorithms and in silico prediction tools have emerged to forecast neoantigens with high immunogenicity, thereby expediting the selection of optimal vaccine candidates [244,248,249]. Moreover, incorporating multi-omics data, which include transcriptomics, proteomics, and epigenomics, offers a more comprehensive understanding of the tumor microenvironment and its influence on the effectiveness of personalized cancer vaccine therapy [250]. These advancements in sequencing and analysis have markedly improved our capacity to develop customized cancer vaccines, and they further emphasize the significance of multidisciplinary collaboration within the cancer immunotherapy field.

### 6.3. Neoantigen Selection

Neoantigen selection is vital for eliciting a robust and effective immune response [20,21]. Factors considered in the selection process include the binding affinity of the neoantigen to major histocompatibility complex (MHC) molecules, the immunogenicity of the epitope, and the likelihood of generating T-cell receptor (TCR) recognition [251,252]. Recent studies have also highlighted the importance of incorporating multi-omics data to ensure that the chosen neoantigens are effectively processed and presented on the cell surface [250]. By combining these innovative approaches, researchers have substantially improved the process of neoantigen selection, bolstering the potential for successful personalized cancer vaccine therapy.

### 6.4. Vaccine Design

Vaccine design directly influences the efficacy of the immune response against tumor cells [8]. Recent research advances have led to the development of various vaccine platforms, including peptide-based, nucleic-acid-based (DNA or RNA), viral-vector-based, and dendritic-cell-based vaccines, each with their own set of advantages and challenges [8,21,22]. The selection of appropriate adjuvants and delivery systems is essential for enhancing the immunogenicity of the vaccine and promoting the activation and expansion of tumor-specific T cells [253]. Innovative techniques such as liposomal and nanoparticle-based delivery systems have shown promise in improving vaccine stability, cellular uptake, and antigen presentation [254]. The swift pace of advancements in vaccine design methodologies, in conjunction with a burgeoning understanding of the tumor microenvironment and immune system, has notably augmented the potential of personalized cancer vaccine therapy, laying the groundwork for more effective and targeted cancer treatments.

### 6.5. Vaccine Production

The high quality of production in personalized cancer vaccine therapy is indispensable for ensuring the delivery of efficacious and safe treatment options to patients [22]. Noteworthy innovations in the production process encompass the incorporation of automation and process optimization to curtail the manufacturing duration and boost scalability [255]. Furthermore, the utilization of continuous manufacturing processes and the establishment of modular facilities have amplified flexibility and adaptability in vaccine production, thereby streamlining the supply chain for personalized therapies [256]. To ensure product quality, regulatory agencies have enforced good manufacturing practices (GMPs) and stringent quality control measures. The integration of advanced bioinformatics tools has also contributed to the acceleration of vaccine development and production, enabling more rapid clinical translation and patient access [110]. As the field of personalized cancer vaccine therapy continues to expand, further innovations in vaccine production technologies and processes will be vital to meeting the growing demand and ensuring the timely delivery of these tailored treatments.

### 6.6. Vaccine Administration

Vaccine administration directly impacts the induction of a robust immune response against tumor cells. The latest research advances have led to the exploration of various routes of administration, including subcutaneous, intradermal, intramuscular, and intranodal, with each route presenting unique benefits and challenges for different vaccine platforms [257,258]. The choice of administration route can influence the vaccine’s biodistribution, antigen presentation, and subsequent immune response [23]. Researchers are also investigating the optimal dosing and scheduling of these personalized vaccines to maximize their efficacy while minimizing potential adverse effects [21,259]. Recent studies have delved into the synergistic effects of merging personalized cancer vaccines with other immunotherapies to amplify therapeutic outcomes and surmount immune resistance [260].

As the realm of personalized cancer vaccine therapy continues to expand, a nuanced understanding of the optimal administration strategies, including route, dosage, and scheduling, will be pivotal for maximizing the therapeutic potential of these individualized treatments.

### 6.7. Immune Response Activation

The activation of the immune response is a central objective in personalized cancer vaccine therapy, aiming to instigate a robust and specific immune response against tumor cells expressing neoantigens. Recent research strides have provided a more comprehensive understanding of the mechanisms underpinning the activation of both innate and adaptive immune responses [20]. Personalized cancer vaccines endeavor to prime the immune system by presenting tumor-specific neoantigens to antigen-presenting cells (APCs) such as dendritic cells, which subsequently activate cytotoxic T lymphocytes (CTLs) to target and eliminate tumor cells [22,178,261]. A key facet of immune response activation lies in the optimization of vaccine design, thereby amplifying the immunogenicity of the vaccine and fostering the expansion of tumor-specific T cells [262].

Recent investigations have probed into combining personalized cancer vaccines with other immunotherapies to augment the antitumor immune response and counter the immune evasion strategies utilized by cancer cells [260]. As the realm of personalized cancer vaccine therapy progresses, a deeper comprehension of immune response activation and its modulation will be indispensable for maximizing the therapeutic potential of these tailored treatments.

### 6.8. Monitoring and Evaluation

Monitoring and evaluation are integral to ensuring the safety, efficacy, and optimization of these individualized treatments. Recent research breakthroughs have resulted in the development of comprehensive methodologies to assess both immune response and clinical outcomes in patients receiving personalized cancer vaccines [21,22]. Essential parameters for evaluating the immunological response include monitoring the expansion of vaccine-specific T cells, the production of cytokines, and the infiltration of immune cells into the tumor microenvironment [258]. These evaluations furnish invaluable insights into the vaccine’s capacity to activate and modulate the immune system.

Clinical evaluation entails tracking objective responses, such as reduction in tumor size, progression-free survival, and overall survival, while also considering the patients’ quality of life [23,263]. As personalized cancer vaccines are often administered in combination with other immunotherapies, it is crucial to identify synergistic effects and ascertain the optimal treatment regimen [260,264].

Moreover, monitoring and evaluating safety profiles is imperative for identifying and managing potential adverse effects associated with personalized cancer vaccines, such as autoimmune reactions or systemic inflammation [109]. The ongoing refinement of monitoring and evaluation strategies will contribute to maximizing the therapeutic potential and safety of personalized cancer vaccine therapies, thereby enhancing patient outcomes.

### 6.9. Follow-Up, Maintenance, and Patient Education and Support

Continuous follow-up empowers healthcare professionals to monitor patients’ responses to therapy, assess potential adverse effects, and make necessary adjustments to treatment plans [21,265]. Maintenance therapy, including administering booster vaccinations or adjusting combination therapies, is pivotal for maintaining the antitumor immune response and preventing cancer recurrence [22,74].

Patient education forms a key aspect of personalized cancer vaccine therapy, as it enables patients to make informed decisions about their treatment and manage potential side effects [257]. Patients should be well informed about potential adverse effects of the treatment and the importance of promptly reporting these to their healthcare providers [109,261]. Moreover, the emotional and psychological wellbeing of patients can significantly influence their response to treatment, underscoring the need for comprehensive support systems. This may include mental health counseling, peer support groups, and resources for family members, creating an environment that nurtures the patient’s resilience and determination [266]. As personalized cancer vaccine therapy continues to progress, it is crucial to maintain an integrated approach that addresses all facets of patient care, including follow-up, maintenance, education, and support. This holistic approach can maximize the therapeutic potential of personalized cancer vaccines and lead to improved patient outcomes [267].

### 6.10. Data Collection and Analysis

Data collection and analysis stand as the cornerstones of personalized cancer vaccine development and evaluation. Recent strides in high-throughput sequencing technologies, computational methodologies, and bioinformatics have ushered in a new era, enabling researchers to more precisely identify and prioritize neoantigens, craft personalized vaccines, and keep track of immune responses [20,22,258]. Furthermore, the application of machine learning algorithms and artificial intelligence has emerged as a powerful tool to navigate the complexity of these data landscapes. This technology bolsters our capacity to predict the immunogenicity of neoantigens and assess the effectiveness of personalized cancer vaccines [252]. Standardized data collection and analysis protocols are paramount to ensure reproducibility, facilitate comparisons across studies, and promote the development of robust and reliable personalized cancer vaccine therapies [23].

### 6.11. Integration with Other Therapies

The integration of personalized cancer vaccines with other therapeutic modalities offers an intriguing avenue in the realm of cancer treatment. By combining different treatment modalities, we can potentially augment therapeutic efficacy, surmount resistance mechanisms, and ultimately improve patient outcomes. Recent scientific findings highlight the potential synergistic effects of marrying personalized cancer vaccines with other forms of immunotherapies. Notably, ICIs, which target immunosuppressive pathways like PD-1 (programmed cell death protein 1), PD-L1 (programmed cell death ligand 1), and CTLA-4 (cytotoxic T-lymphocyte-associated protein 4), have shown promising results in combination with personalized cancer vaccines [109]. The rationale behind this combination is that the vaccine can stimulate a specific antitumor immune response, while ICIs can further enhance the function and persistence of tumor-specific T cells [191].

Additionally, personalized cancer vaccines can be combined with conventional therapies, such as chemotherapy and radiotherapy, to induce immunogenic cell death and release tumor-associated antigens, creating a more conducive environment for the activation of vaccine-induced immune responses [139]. Furthermore, combining personalized cancer vaccines with targeted therapies, such as kinase inhibitors or monoclonal antibodies, has shown promise in preclinical models by modulating the tumor microenvironment and improving immune cell infiltration [265,268].

### 6.12. Expanding Applications

The exploration of personalized cancer vaccine therapy applications continues to surge, unlocking new possibilities for cancer treatment across a diverse range of tumor types and stages. Groundbreaking studies have revealed the potential and effectiveness of tailored cancer vaccines against melanoma, glioblastoma, and non-small-cell lung cancer [22,263]. Aside from solid tumors, there is growing evidence of personalized cancer vaccines proving promising against hematological malignancies such as acute myeloid leukemia (AML) and multiple myeloma (MM) [269,270]. Furthermore, researchers are delving into the possibility of implementing personalized cancer vaccines during earlier disease stages, or as supplementary therapy after surgery or radiotherapy, with the ultimate goal of preventing relapse or disease progression [126].

As the development of personalized cancer vaccine therapy progresses, gaining a comprehensive understanding of the factors impacting vaccine effectiveness—including the tumor microenvironment, individual immune responses, and the dynamic interplay among various treatment approaches—becomes essential in extending its application scope. Unrelenting research and innovation are indispensable for harnessing the full therapeutic capacity of personalized cancer vaccines and enhancing outcomes for a wide spectrum of cancer patients. The world of clinical trials is abuzz with personalized cancer vaccines and neoantigen-targeting therapies [271,272]. Challenges exist, such as optimizing vaccine design and manufacturing, plus pinpointing the patients most likely to benefit from personalized immunotherapies [195,273]. Add in resistance mechanisms, sky-high costs, and complex production, and suddenly these groundbreaking treatments seem less accessible and affordable. So, the cancer immunotherapy research needs to zero in on resistance, refine neoantigen identification tech, and make personalized therapies more cost-effective and scalable [274]. This is where multi-omics approaches and AI come into play, helping to precisely stratify patients and craft combination therapies that boost efficacy while minimizing side effects [75]. Tackling these challenges and embracing emerging technologies will be the key to revolutionizing cancer treatment and offering new hope to countless patients.

## 7. Conclusions

The future prospects of tumor vaccines combined with other treatment modalities are compelling, driven by advances in our understanding of the tumor microenvironment and the immune system’s interaction with cancer [275]. Enhanced delivery and efficacy of tumor vaccines could be achievable with the development of nanotechnology and advanced drug delivery systems, which increase the stability and targeting of tumor antigens and immune adjuvants, enhancing their immunogenicity when combined with other treatments [276]. Another promising avenue is the combination of tumor vaccines with adoptive cellular therapies such as CAR-T cells, where the vaccine’s ability to induce a broad immune response could complement the direct cytotoxic effect of engineered T cells [277]. Continual clinical trials and progressive insights into cancer immunology underline the burgeoning promise of tumor vaccines in concert with other treatments [278].

Personalized cancer vaccines are another exciting frontier in cancer therapy, capitalizing on advancements in our understanding of the immune system and cancer genomics to deliver highly individualized treatments. The development of predictive biomarkers and novel imaging techniques may enhance the efficacy of personalized cancer vaccines by facilitating real-time monitoring of patient responses, enabling timely adjustments to treatment strategies [279]. Rapid progress in genomics, bioinformatics, and manufacturing technologies has streamlined the process of identifying neoantigens and producing personalized vaccines, potentially making these treatments available to a larger number of patients [280]. With continued research and technological innovation, personalized vaccines potentially signify a significant breakthrough in cancer treatment [281].

Meanwhile, we must acknowledge significant persisting challenges, such as the high cost and logistical complexity of producing individualized vaccines, the necessity for more comprehensive clinical trials to thoroughly appraise their effectiveness, and the fact that not all patients’ tumors harbor identifiable neoantigens [258]. Furthermore, potential issues such as determining the optimal timing and sequencing of combination therapies, managing possible side effects and immune-related toxicities, and addressing issues of cost and accessibility cannot be overlooked [197]. Despite these hurdles, the potential for leveraging the inherent capabilities of the immune system in combatting tumors is distinctly promising, hinting at a future marked by enhanced patient prognostic outcomes.

## Figures and Tables

**Figure 1 pharmaceuticals-16-01384-f001:**
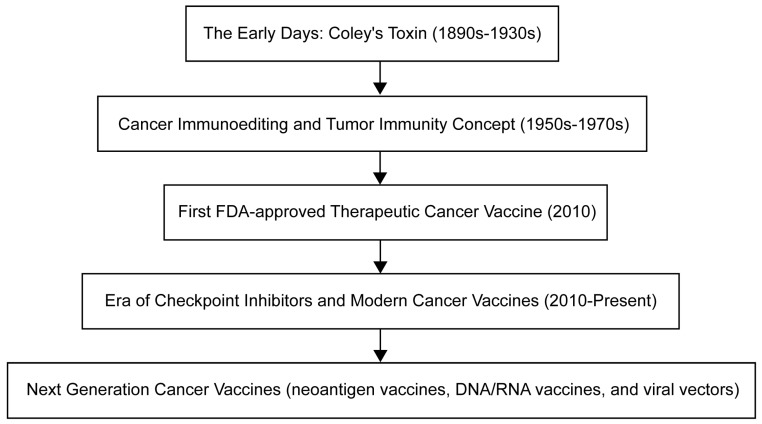
The timeline of tumor vaccines’ development.

**Figure 2 pharmaceuticals-16-01384-f002:**
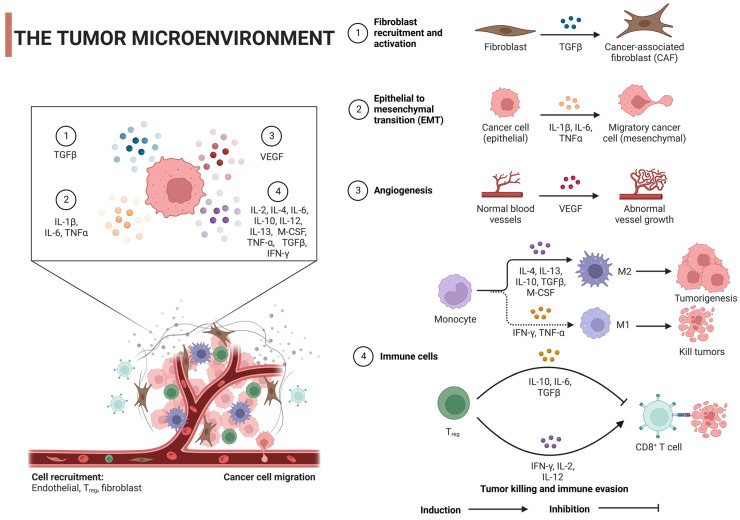
The regulation mechanisms of the tumor microenvironment on tumor cells: TGF-β, transforming growth factor-beta; IL, interleukin; VEGF, vascular endothelial growth factor; TNF-α, tumor necrosis factor-α; IFN-γ, interferon-γ; M-CSF, macrophage colony-stimulating factor; Tregs, regulatory T cells. Created with BioRender.com.

**Figure 3 pharmaceuticals-16-01384-f003:**
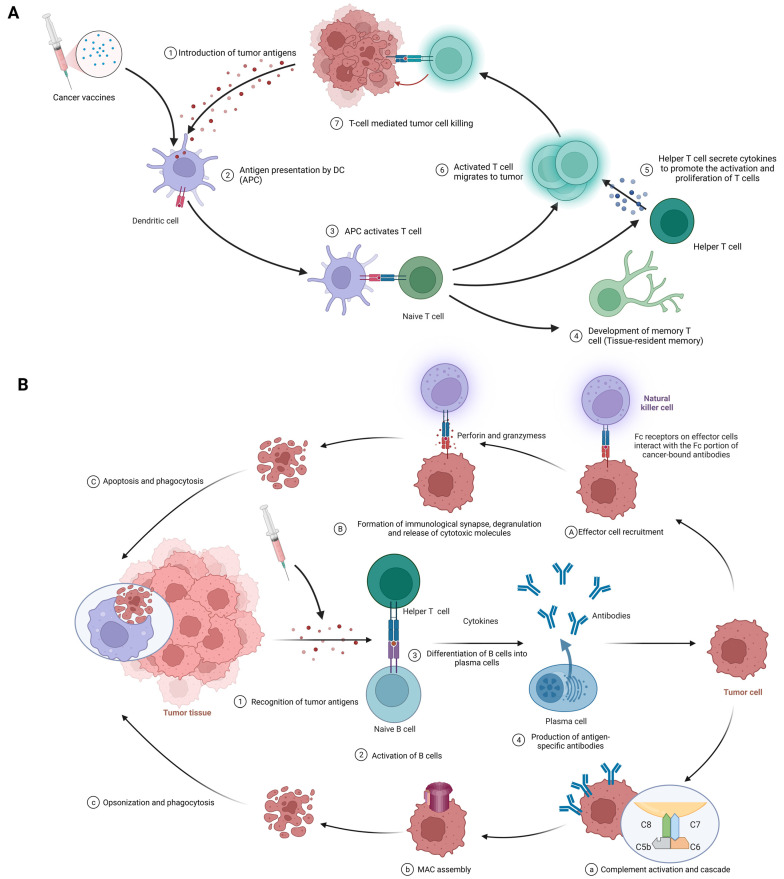
(**A**) Cellular immune response against cancer (step 1 to step 7); APCs, antigen-presenting cells; APC, antigen-presenting cell. (**B**) Humoral immunity response against cancer (step 1 to step 4). A–C; antibody-dependent cell-mediated cytotoxicity; a–c: complement-dependent cytotoxicity; MAC, membrane attack complex. Created with BioRender.com.

**Figure 4 pharmaceuticals-16-01384-f004:**
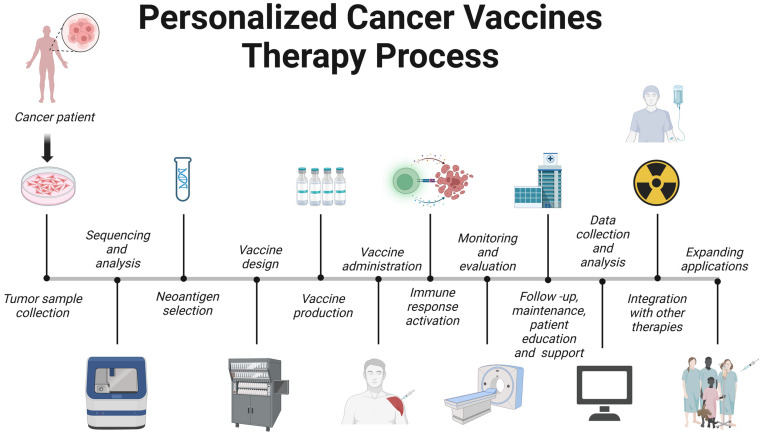
Therapeutic pipeline for personalized vaccines. Created with BioRender.com.

## Data Availability

Data sharing is not applicable..

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
