# Peer review of "Tumor Vaccines: Unleashing the Power of the Immune System to Fight Cancer"

_pharmaceuticals, 2023, doi:10.3390/ph16101384_

Round 1

Reviewer 1 Report

This review manuscript well summarized current frontiers on the cancer vaccine development, including the mechanism of vaccination, different types of vaccine (peptide-based, nucleic acid-based, viral vector-based, and cell-based), the combination therapy of cancer vaccine with other type of therapies.  

The review is well-organized and very informative, I suggest accept in current form.With that, I do think that the figures are in low resolution. I'm not sure if it's a formatting issue for the images are low quality originally. It would be much better if the figures have higher resolution.

There are some informal use of English (line 375, antigen selection is "supper" important, I think there are more formal choices of words to replace "super", same in line 378)  

Overall I think that this manuscript (Tumor Vaccines: Unleashing the Power of the Immune System to Fight Cancer) is a review article rather than a research article.  

This review manuscript comprehensively covers the topic of tumor vaccines: the mechanism of tumor vaccines, the characteristics of tumor microenvironment and how it affect the application of immunotherapies, the different types of tumor vaccines and their respective advantages and disadvantages, current cancer vaccines in the pharmaceutical pipelines, combination therapies of tumor vaccine with other types of therapies, and the design and workflow of future personalized cancer vaccines.  

When mentioning the dendritic cell-based vaccines, there are and very recent and relevant work that's worth being included in this review (https://www.nature.com/articles/s41467-023-40886-7)

The authors should also mention in situ cancer vaccines that can circumvent the need to identify the accurate antigen.  

Overall this review gives comprehensive information on cancer vaccines, which is a big topic in the current cancer-related research field. 

The content is very informative and relevant to the journal. Improvements can be made at:

1. Use higher resolution figures,

2. Include the aforementioned work to make the review more well-rounded, and

3. Having a section to discuss Table 1 in detail (what are the components, what are the target disease, what are the stages of development, etc.)

Author Response

Dear Reviewer:

I would like to extend my profound gratitude for the time, effort, and expertise you have invested in reviewing my manuscript. Your constructive feedback and insightful comments have been instrumental in enhancing the quality and clarity of the work. We appreciate your reviewing and providing us with a high-quality review report for our manuscript entitled " Tumor Vaccines: Unleashing the Power of the Immune System to Fight Cancer "(Manuscript ID: 2617762). These comments and suggestions are valuable to us and will help us to improve the quality of the manuscript.

I earnestly appreciate your continued guidance and support as I navigate this scholarly journey. Your role in refining the intellectual fabric of this work cannot be understated. It's my genuine hope that my responses and revisions will align with the academic standards and expectations you hold for the manuscript. We've read your review report in detail. Additionally, we've modified the relevant manuscript parts based on your review comments, so please check again whether our modifications are appropriate.

Our responses to your comments are as follows:

Comment 1:

“I do think that the figures are in low resolution. I'm not sure if it's a formatting issue for the images are low quality originally. It would be much better if the figures have higher resolution.”                  

Response:

We appreciate your recommendations concerning the graphical content of the manuscript. We concur with your perspectives. All images have been meticulously refined to enhance clarity and align with the journal's specifications, and we trust they will align with your scholarly standards.

Comment 2:

“There are some informal use of English (line 375, antigen selection is "supper" important, I think there are more formal choices of words to replace "super", same in line 378)”

Response:

we are grateful for pointing out the informal use of language in certain lines. This have been corrected to maintain the formal and scientific tone of the manuscript (Page13-14, Line491-495):

“Antigen selection is of paramount significance. The antigens should be specific to the tumor or associated with it, so the immune response zeroes in on cancer cells without harming healthy tissues. Plus, the chosen antigens need to be highly immunogenic and able to stimulate both CD8+ cytotoxic T cells and CD4+ helper T cells for a strong, long-lasting attack against tumors.”

Comment 3:

“When mentioning the dendritic cell-based vaccines, there are and very recent and relevant work that's worth being included in this review (https://www.nature.com/articles/s41467-023-40886-7)”

Response:

We recognize the significance of the recent work on dendritic cell-based vaccines. We've incorporated this reference to provide a more well-rounded view on the topic (Page11, Line426-432):

“In recent study, researchers have introduced a metabolic glycan labeling technique using azido-sugars for the enhancement of DC vaccines(1). This method not only boosts DC activation and antigen presentation but also facilitates the efficient conjugation of cytokines(1). Furthermore, it holds promise for broad applications across various tumors, provides a platform for modulating interactions between DCs and other immune cells, and amplifies the antitumor efficacy of dendritic cell vaccines.”

  1. Han J, Bhatta R, Liu Y, Bo Y, Elosegui-Artola A, Wang H. Metabolic glycan labeling immobilizes dendritic cell membrane and enhances antitumor efficacy of dendritic cell vaccine. Nat Commun. 2023;14(1):5049. (Page30, Line1198-1199)

Comment 4:

“The authors should also mention in situ cancer vaccines that can circumvent the need to identify the accurate antigen.”

Response:

The suggestion to delve deeper into in situ cancer vaccines and the importance of identifying accurate antigens will certainly enrich the manuscript. This has been added in chapters 4.6 (Page13, Line476-485):

4.6. Another cancer vaccine therapy: In situ cancer vaccines

In situ cancer vaccines represent a therapeutic approach where the tumor inside a patient's body is directly targeted to serve as its own vaccine. Rather than extracting tumor cells for external processing and reintroduction, in situ vaccines stimulate the immune system by damaging the tumor in its native environment. As the tumor cells die, they release antigens, which are then recognized by the immune system. Often, this is achieved by injecting immune-stimulating agents or oncolytic viruses into the tumor. This not only aims to destroy the immediate tumor but also primes the immune system to recognize and combat tumor cells elsewhere in the body(176). In situ cancer vaccines show promise in preliminary studies.”

Comment 5:

“Having a section to discuss Table 1 in detail (what are the components, what are the target disease, what are the stages of development, etc.)”

Response:

We concur with your astute recommendation emphasizing the imperative of a comprehensive analysis of Table 1 to delineate the ingredients, target diseases, and stages of development clearly. We've meticulously integrated this discussion into the introductions of each vaccine category, namely peptide-based, DNA/RNA-based, viral vector-based, dendritic cell-based, and whole cell-based vaccines, and anticipate that this enhancement meets your scholarly expectations:

“At present, the relatively mature peptide vaccines include:Nelipepimut-S (NeuVax), CIMAvax-EGF and MUC1-based peptide vaccine. Nelipepimut-S, also known as NeuVax, is a peptide vaccine targeting HER2/neu-expressing cancer cells, primarily focusing on early-stage HER2 1+ and 2+ breast cancer patients ineligible for standard HER2 therapies. It combines the E75 peptide from HER2/neu with GM-CSF adjuvant for heightened immune response. Although its scope may cover other HER2/neu cancers like ovarian and gastric, its clinical development hit snags. The Phase III present trial was halted due to endpoint concerns, though research persists on its synergistic potentials. Conversely, CIMAvax-EGF, targeting the Epidermal Growth Factor for NSCLC, merges recombinant EGF with a protein carrier. It's shown promise in prolonging the lives of late-stage lung cancer patients, with Phase III trials completed and further studies abroad. Additionally, MUC1-based peptide vaccines focus on the aberrantly expressed glycoprotein in cancers like breast and pancreatic. Though some have reached Phase I and II trials with promising safety and immune indicators, eliciting robust clinical responses remains complex, leading to exploration in combination therapies. Collectively, these vaccines represent cutting-edge cancer treatment approaches, each with its unique targets and development stages.” (Page9, Line324-340)

“CV9104 is an mRNA-based cancer vaccine targeting prostate cancer. Its development reached a Phase II trial for metastatic castration-resistant prostate cancer (mCRPC). This vaccine represents the innovative utilization of mRNA in oncology.” (Page10, Line370-373)

“OncoVEXGM-CSF, or T-VEC, is an oncolytic HSV-1 vaccine modified for tumor selectivity and GM-CSF production, primarily targeting melanoma. It gained FDA approval for unresectable recurrent melanoma post a successful Phase III trial. CG0070, another adenovirus-based vaccine, is engineered for selective replication in Rb pathway-defective cancer cells and targets bladder cancer. LV305, a lentivirus-based vaccine, delivers the NY-ESO-1 antigen gene to dendritic cells, aiming at NY-ESO-1 expressing cancers like melanoma and sarcoma. JX-594 or Pexa-Vec, a vaccinia virus-based vaccine, is modified to express GM-CSF and selectively target cancer cells with high thymidine kinase activity and it underwent several trials, including a Phase III for hepatocellular carcinoma. These represent innovative intersections of virotherapy and immunotherapy in oncology.” (Page10, Line401-411)

“Outstanding representatives include: Provenge and DCVax-L. Provenge (Sipuleucel-T) is an FDA-approved autologous cellular immunotherapy for advanced prostate cancer. It uses a patient's peripheral blood mononuclear cells (PBMCs), exposed to a fusion protein, PA2024, which combines an antigen from prostate cancer cells with an immune activator, GM-CSF, priming an immune response against prostate cancer cells expressing the antigen. On the other hand, DCVax-L is an autologous dendritic cell vaccine for glioblastoma multiforme (GBM). The vaccine is prepared by loading a patient's dendritic cells with tumor lysate from their own tumor tissue, enabling the immune system to recognize and attack corresponding cancer cells. Both vaccines harness dendritic cells to target cancer, but their clinical journeys and disease targets differ.” (Page11, Line433-442)

“Representatives include: GVAX, Canvaxin and Oncophage. GVAX is a whole-cell tumor vaccine, utilizing tumor cells genetically modified to secrete GM-CSF, an immune stimulant, and has been explored for cancers like pancreatic and prostate, with mixed outcomes in later-phase trials. Canvaxin, aimed at melanoma, combines irradiated autologous and allogeneic melanoma cells with the BCG adjuvant, but failed to show significant survival benefits in a Phase III trial for advanced melanoma. Oncophage (vitespen) is derived from patient-specific tumor heat shock proteins (HSP) and targets primarily renal cell carcinoma and melanoma. It completed Phase III trials with mixed results but secured approval in Russia for kidney cancer treatment. While these vaccines showcase varied cancer immunotherapy strategies, each has faced challenges in late-stage clinical evaluations.” (Page11-12, Line461-471)

Warm Regards,

Guangzhen Wu

Reviewer 2 Report

The Review article provides a comprehensive overview of cancer vaccine therapy, covering various aspects of the treatment process, versatile vaccine types, recent research advancements, and challenges associated with this innovative approach. The paper presents a well-structured and informative discussion, but there are a few points to address and clarify.

1.     Section 2 mentions that alterations in the extracellular matrix (ECM) can influence angiogenesis, immune evasion, and therapy resistance. Can you provide examples of specific ECM alterations and their consequences in the context of cancer progression?

2.     The discussion of immune cells within the TME and their dual roles in either promoting or inhibiting tumor progression is insightful. However, could you elaborate on the factors or signals that determine whether immune cells in the TME have pro-tumor or anti-tumor effects?

3.     The section 2 mentions several cytokines present in the TME and their effects on tumor growth, survival, and immune responses. It would be valuable to discuss the potential therapeutic implications of targeting these cytokines in cancer treatment.

4.     The section also hints at the relevance of the TME to cancer vaccine efficacy. Could you expand on how tumor vaccines interact with the TME and the strategies researchers are exploring to modulate the TME to enhance the effectiveness of cancer vaccines?

5.     Could the authors create a fresh table that outlines the strengths and weaknesses of each vaccine method as elaborated upon in section 3?

6.     Section 4.2, the section mentions challenges such as the efficient delivery of DNA/RNA into cells and the risk of autoimmune responses. Are there specific techniques or technologies being developed to improve the delivery of genetic material in these vaccines while minimizing associated risks?

7.     Section 4.3, the challenges related to pre-existing immunity to viral vectors are mentioned. Could you elaborate on strategies being explored to mitigate these challenges and enhance the efficacy of viral vector-based tumor vaccines?

8.     The section 4.5 “Influencing factors of tumor vaccines” should be section 4.6. In this section, the role of adjuvants and delivery systems is highlighted. Can you elaborate on any recent breakthroughs or innovative approaches in adjuvant development that hold potential for enhancing cancer vaccine efficacy?

9.     Section 5.1, it's mentioned that this combination generates tumor-specific T cells and prevents their exhaustion. Could you elaborate on the mechanisms behind these effects?

10.  Section 5.2, can authors provide examples of chemotherapeutic agents that have shown promise in combination with cancer vaccines?

11.  Section 5.3, how does radiotherapy affect the release of DAMPs and tumor vessel permeability in the tumor microenvironment? Could you provide more details on the mechanisms involved?

12.  Section 5.5, the dual-action mechanism of oncolytic virotherapy is intriguing. Can you elaborate on how this approach stimulates the immune system in response to the viral infection? What are some recent advancements in oncolytic virotherapy that have improved its effectiveness or safety?

No concerns about the quality of language.

Author Response

Dear Reviewer:

I would like to extend my profound gratitude for the time, effort, and expertise you have invested in reviewing my manuscript. Your constructive feedback and insightful comments have been instrumental in enhancing the quality and clarity of the work. We appreciate your reviewing and providing us with a high-quality review report for our manuscript entitled " Tumor Vaccines: Unleashing the Power of the Immune System to Fight Cancer "(Manuscript ID: 2617762). These comments and suggestions are valuable to us and will help us to improve the quality of the manuscript.

I earnestly appreciate your continued guidance and support as I navigate this scholarly journey. Your role in refining the intellectual fabric of this work cannot be understated. It's my genuine hope that my responses and revisions will align with the academic standards and expectations you hold for the manuscript. We've read your review report in detail. Additionally, we've modified the relevant manuscript parts based on your review comments, so please check again whether our modifications are appropriate.

Our responses to your comments are as follows:

Comment 1:

“Section 2 mentions that alterations in the extracellular matrix (ECM) can influence angiogenesis, immune evasion, and therapy resistance. Can you provide examples of specific ECM alterations and their consequences in the context of cancer progression?”                  

Response:

We've delved deeper into specific ECM alterations and their direct consequences on cancer progression, providing tangible examples to elucidate this further (Page3, Line106-127):

“The ECM, a tangled web of proteins, glycoproteins, and proteoglycans, lends structure and shapes cell behavior via biochemical and biomechanical signals. When cancer messes with ECM composition, it can boost angiogenesis, immune evasion, and therapy resistance. For example, increased ECM stiffness, as seen with lysyl oxidase (LOX) overexpression, promotes cellular proliferation and survival through mechanisms like focal adhesion kinase (FAK) activation, while also enhancing tumor invasiveness. Similarly, an accumulation of hyaluronan, often linked to poor prognosis, can bolster tumor growth and facilitate immune evasion. Altered expression of matrix metalloproteinases (MMPs) can remodel the ECM, favoring tumor invasion by releasing growth factors. Furthermore, specific ECM proteins like elastin, laminin, tenascin-C and periostin, when overexpressed, support tumor cell migration and survival. Notably, changes in collagen orientation, resulting in aligned fibers, provide pathways for enhanced tumor cell migration, with such alignment often indicating a higher risk of metastasis. ECM is also involved in the secretion of various growth factors, like transforming growth factor-β (TGF-β), Interleukin-1β (IL-1β), IL-6, Tumor necrosis factor α (TNF-α), and vascular endothelial growth factor (VEGF) are secreted by various cells of the TME and they can initiate tumor cell growth, survival, migration, angiogenesis ,and epithelial–mesenchymal transition (EMT). This is achieved by regulating their specific receptors and stimulating signaling pathways (Figure 2). This intricate interplay between the ECM and tumor cells not only propels cancer progression but also presents challenges in therapy due to factors like drug penetration barriers and the activation of cellular survival pathways.”

Comment 2:

“The discussion of immune cells within the TME and their dual roles in either promoting or inhibiting tumor progression is insightful. However, could you elaborate on the factors or signals that determine whether immune cells in the TME have pro-tumor or anti-tumor effects?”

Response:

Your observation regarding the factors or signals influencing the orientation of immune cells within the TME toward either pro-tumor or anti-tumor activities is astute. We shall provide a more comprehensive discussion (Page4, Line128-138):

“Similarly, immune cell functions shaped by a blend of elements. Tumor cells can secrete immunosuppressive cytokines and checkpoint ligands that modify the immune response, while conditions like hypoxia and an acidic pH, resulting from metabolic changes in tumors, can suppress immune activity. Concurrently, metabolic competition due to glucose consumption by tumors, alterations in the extracellular matrix, and the recruitment of immunosuppressive cells can inhibit effective immune responses. Direct cell-cell interactions within the TME, the gut microbiome's influence on tumor immunity, therapeutic interventions, and the presence of chronic inflammation further modulate the balance between pro-tumor and anti-tumor effects. The interplay of these factors determines the complex and dynamic nature of immune responses within the TME.”

Comment 3:

“The section 2 mentions several cytokines present in the TME and their effects on tumor growth, survival, and immune responses. It would be valuable to discuss the potential therapeutic implications of targeting these cytokines in cancer treatment.”

Response:

We've discussed in greater depth the therapeutic implications of targeting the mentioned cytokines, emphasizing their potential role in revolutionizing cancer treatment (Page4-5, Line179-188):

“TME offers promising therapeutic avenues in cancer treatment. Immunosuppressive cytokines like TGF-β, when inhibited, may restore anti-tumor immune responses and curtail metastasis. Although IL-10 generally suppresses TME immunity, its nuanced roles suggest potential benefits from modulating its levels, while the inhibition of angiogenic and immunosuppressive VEGF has birthed FDA-approved therapies like bevacizumab. On the immune-stimulatory front, IL-2, known to bolster T cell growth, has seen therapeutic applications, albeit with side effects, and IL-12's potent activation of immune cells hints at its combinatorial therapeutic potential. Additionally, checkpoint inhibitors targeting the PD-1/PD-L1 axis, such as pembrolizumab, rejuvenate exhausted T cells to counteract tumors.”

Comment 4:

“The section also hints at the relevance of the TME to cancer vaccine efficacy. Could you expand on how tumor vaccines interact with the TME and the strategies researchers are exploring to modulate the TME to enhance the effectiveness of cancer vaccines?”

Response:

The interaction of tumor vaccines with the TME and potential strategies to modulate the TME for enhanced vaccine efficacy have been elaborated upon, addressing the connection and its clinical implications. (Page5, Line189-199):

“Together, TME significantly impacts the success of cancer vaccines. While vaccines aim to activate immune cells against tumor antigens, the TME's immunosuppressive nature can stymie these activated T cells. Factors like altered tumor antigen presentation, reduced MHC molecule expression, and immune checkpoint expression further hinder vaccine-induced responses. To enhance cancer vaccine efficacy, researchers are exploring combined therapies with checkpoint inhibitors, methods to reduce immunosuppressive cells in the TME, strategies to breach the TME's physical barriers, the incorporation of potent adjuvants, cytokine modulation, and the development of personalized vaccines tailored to individual tumor antigen profiles. These multi-pronged strategies, targeting both the vaccine mechanism and the TME, are steering the direction of next-generation cancer therapies.”

Comment 5:

“Could the authors create a fresh table that outlines the strengths and weaknesses of each vaccine method as elaborated upon in section 3”

Response:

We concur with your recommendation. To maintain the article's coherence without an overabundance of tables, we've incorporated a synthesis of the advantages and disadvantages of each vaccine approach within Table 1 for a streamlined comparison (Page12-13, Line472-475):

Table 1. Below is a tabular list of various tumor vaccines in the last decade.

Types of tumor vaccines

Strengths

Weaknesses

Peptide Vaccines

1.       Specific to tumor antigens

2.       Low toxicity

3.       Easily synthesized and scalable

1.       Limited to known antigens

2.       May not induce a robust immune response alone

DNA/RNA-based Vaccines

1.       Can encode multiple antigens

2.       Flexibility in design

3.       Stable and easy to produce

1.       Delivery into cells can be challenging

2.       Risk of integration (for DNA)

3.       May induce autoimmune responses

Viral vector-based Vaccines

1.       Efficient cell entry and expression

2.       Can induce strong immune responses

1.       Pre-existing immunity to the viral vector can reduce effectiveness

2.       Potential for off-target effects

Dendritic cell-based Vaccines

1.       Tailored to individual patients

2.       Induce potent T-cell responses

1.       Labor-intensive and costly production

2.       Requires patient-specific infrastructure

Whole cell-based vaccines

1.       Broad range of tumor antigens presented

2.       Mimics natural infection

1.       Complex manufacturing

2.       Potential for tumor cell growth if not fully inactivated

The preceding table encapsulates seminal instances of assorted classifications of cancer vaccines, encompassing peptide-based, DNA/RNA-based, viral vector-based, dendritic cell-based vaccines, along with whole cell-based vaccines. Each paradigm is delineated in exhaustive detail, supplemented by pertinent bibliographical citations for subsequent scholarly inquiry.

Comment 6:

“Section 4.2, the section mentions challenges such as the efficient delivery of DNA/RNA into cells and the risk of autoimmune responses. Are there specific techniques or technologies being developed to improve the delivery of genetic material in these vaccines while minimizing associated risks?”                  

Response:

We've elaborated on emerging techniques and technologies aimed at enhancing the delivery of genetic material in vaccines and the mitigation of potential risks (Page9-10, 353-368):

“However, recent advances, including several vaccines in clinical trials and technological progress enhancing vaccine delivery efficiency, are encouraging. The field looks forward to leveraging advancements in genomic sequencing, bioinformatics, and nanotechnology to surmount current limitations, aspiring to blend these potent vaccines with other immunotherapeutic strategies for comprehensive cancer eradication. Among them, techniques such as lipid nanoparticles (LNPs), which have been seminal for mRNA COVID-19 vaccines, are being adapted for cancer vaccine development to boost the delivery of tumor-specific antigens. Electroporation and viral vectors, like adenoviruses, enhance the uptake of DNA/RNA, while non-viral nanocarriers and micro-needle patches aim to augment this delivery without inducing strong anti-vector responses. To mitigate autoimmune risks, researchers emphasize tumor-specific antigen selection, sequence optimization to reduce cross-reactivity, and transient expression techniques, such as those inherent to mRNA vaccines. Furthermore, tolerance-breaking adjuvants and nanoparticles tailored for targeted delivery are being harnessed to fine-tune the immune response, maximizing anti-tumor efficacy while minimizing collateral damage to healthy tissues.”

Comment 7:

“Section 4.3, the challenges related to pre-existing immunity to viral vectors are mentioned. Could you elaborate on strategies being explored to mitigate these challenges and enhance the efficacy of viral vector-based tumor vaccines?”

Response:

Strategies to counter the challenges posed by pre-existing immunity to viral vectors have been expanded upon, offering insights into the latest research and innovations in this domain (Page10, Line389-400):

“To counteract this, researchers are exploring a range of strategies: using rare or novel viral vectors with limited human exposure, pseudotyping to change viral envelope proteins, employing a heterologous prime-boost strategy with different vectors, making genetic modifications to the viral capsid to reduce recognizability, co-administering with immune modulators to transiently suppress certain immune responses, opting for non-intravenous delivery routes like intratumoral administration to avoid high antibody concentrations, and adjusting dosages, either by using a high vector dose to overcome neutralization or administering repeated low doses to evade immune detection. Additionally, adjuvants are being explored to shift the immune response focus from the vector to the delivered tumor antigen. These multifaceted approaches aim to optimize the efficacy of viral vector-based tumor vaccines in the face of pre-existing immunity.”

Comment 8:

“The section 4.5 “Influencing factors of tumor vaccines” should be section 4.6. In this section, the role of adjuvants and delivery systems is highlighted. Can you elaborate on any recent breakthroughs or innovative approaches in adjuvant development that hold potential for enhancing cancer vaccine efficacy?”

Response:

We've delved into recent breakthroughs and innovations in adjuvant development and delivery systems, underscoring their potential significance in enhancing cancer vaccine efficacy (Page13-14, Line487-510):

“Boosting the power of cancer vaccines is a top priority for researchers. They're diving deep into adjuvants and combination therapies to ramp up immune responses, outsmart tumor immune evasion, and prevent cancer from coming back. Efficacy of cancer vaccines hinges on several factors: Picking the right antigens, choosing adjuvants wisely, and using the best delivery systems. Antigen selection is of paramount significance. The antigens should be specific to the tumor or associated with it, so the immune response zeroes in on cancer cells without harming healthy tissues. Plus, the chosen antigens need to be highly immunogenic and able to stimulate both CD8+ cytotoxic T cells and CD4+ helper T cells for a strong, long-lasting attack against tumors. Adjuvants help by making cancer vaccines more immunogenic. They stimulate the innate immune system, encourage antigen uptake by APCs, and help activate and expand antigen-specific T cells. There are different types of adjuvants, like alum, toll-like receptor (TLR) agonists, and cytokines, each with unique action mechanisms and varying effectiveness. The latest research points out, TLR agonists, such as TLR9 and TLR7/8 agonists, have shown promise by bolstering antigen-presenting cells and intensifying immune responses. Researchers have developed a nanosystem that can inhibit a process called MerTK-mediated efferocytosis. This inhibition leads to the release of immunogenic contents into the tumor microenvironment, potentially boosting the body's natural defenses against the tumor. Similarly, STING agonists enhance dendritic cell activity, boosting T-cell responses against tumors. Oncolytic viruses, while serving as direct anti-tumor agents, also act as adjuvants by releasing tumor antigens within an inflammatory milieu. These recent breakthroughs encapsulate the dynamic progression in adjuvant research, aiming to optimize the immune system's potency against tumors.”

Comment 9:

“Section 5.1, it's mentioned that this combination generates tumor-specific T cells and prevents their exhaustion. Could you elaborate on the mechanisms behind these effects?”

Response:

The mechanisms behind the generation of tumor-specific T cells and their prevention of exhaustion have been detailed further to provide a clear understanding (Page14, Line530-540):

“The combined approach has shown promise in preclinical models and early clinical trials by generating tumor-specific T cells and preventing their exhaustion. The mechanisms behind these effects is that cancer vaccines aim to boost T cell recognition of tumor antigens, but this immune response can be dampened by the tumor's evasion mechanisms. Enter immune checkpoint inhibitors, which block inhibitory checkpoints (PD-1 and CTLA-4) on T cells, essentially "releasing the brakes" and amplifying their anti-tumor activity. By combining cancer vaccines, which enhance the number of tumor-recognizing T cells, with checkpoint inhibitors that ensure these T cells are not suppressed, there's a synergistic boost in the anti-tumor immune response. Preliminary studies suggest this combination augments tumor attack, potentially leading to improved patient outcomes.”

Comment 10:

“Section 5.2, can authors provide examples of chemotherapeutic agents that have shown promise in combination with cancer vaccines?”

Response:

We've provided examples of chemotherapeutic agents that have shown potential synergistic effects when combined with cancer vaccines. (Page15, Line547-557):

“Several chemotherapeutic agents have been identified to potentially enhance the efficacy of cancer vaccines due to their immune-modulating effects. For instance, cyclophosphamide and temozolomide can deplete immune-suppressing regulatory T cells (Tregs), creating a more receptive tumor environment for vaccine action. Docetaxel, used for cancers like breast and prostate, can bolster antigen presentation, thereby enhancing immune recognition of tumor cells. Gemcitabine targets and reduces myeloid-derived suppressor cells (MDSCs). When combined with cancer vaccines, these agents can modify the tumor environment, diminish immune suppression, or amplify the immune response against tumors, though the combination's choice depends on multiple factors, including cancer type and patient health.”

Comment 11:

“Section 5.3, how does radiotherapy affect the release of DAMPs and tumor vessel permeability in the tumor microenvironment? Could you provide more details on the mechanisms involved?”

Response:

The effects of radiotherapy on the release of DAMPs and tumor vessel permeability has been detailed, offering clarity on the underlying mechanisms (Page15, Line575-583):

“Research indicates that radiotherapy exerts both cytotoxic and immunomodulatory effects on the tumor microenvironment. Beyond directly damaging tumor cells, RT induces immunogenic cell death, leading to the release of damage-associated molecular patterns (DAMPs). These DAMPs serve as "danger signals", enhancing dendritic cell function and fostering anti-tumor immune responses. Concurrently, Radiotherapy damages the tumor vasculature, increasing its permeability due to direct effects on endothelial cells and the upregulated release of VEGF from irradiated tumor cells. This can lead to both transient improvement in oxygen and nutrient delivery and enhanced immune cell infiltration into the tumor.”

Comment 12:

“Section 5.5, the dual-action mechanism of oncolytic virotherapy is intriguing. Can you elaborate on how this approach stimulates the immune system in response to the viral infection? What are some recent advancements in oncolytic virotherapy that have improved its effectiveness or safety?”

Response:

The intriguing dual-action mechanism of oncolytic virotherapy has been elaborated upon, including the latest advancements in its efficacy and safety parameters:

“This dual-action mechanism that harmonizes direct cellular destruction with immune activation embodies a promising pathway in the realm of cancer therapy. The dual-action mechanism encompasses direct tumor cell lysis, releasing tumor-associated antigens, and the unveiling of damage-associated molecular patterns (DAMPs) and pathogen-associated molecular patterns (PAMPs). These elements activate both the innate and adaptive arms of the immune system, enhancing anti-tumor responses. As newly assembled viral entities continue their onslaught on other malignant cells, a self-propagating cycle is established.” (Page16, Line620-627)

“Cancer vaccines and oncolytic virotherapy offer a potential synergistic approach for cancer treatment. Cancer vaccines introduce cancer-specific antigens into the body, training the immune system to recognize and attack cells displaying these antigens. Concurrently, oncolytic virotherapy uses engineered viruses that selectively infect and eliminate cancer cells, subsequently releasing tumor antigens and new viral particles that can infect nearby cancer cells, thereby stimulating an immune response. In combination, the cancer vaccine's potential enhancement of the immune response, coupled with the direct cellular damage from the oncolytic virus, could increase therapeutic effectiveness. Initial studies suggest this combination can result in a more robust and long-lasting immune response against tumors, even potentially overcoming some immune evasion tactics employed by cancer cells. Nevertheless, this approach is not without limitations. The immune system could respond to the oncolytic virus, reducing its cancer-killing effectiveness. Additionally, delivering both oncolytic viruses and cancer vaccines to the tumor site, particularly in solid tumors, is challenging. The diverse nature of cancers and variability in patient responses can limit the overall responsiveness of this combined therapy. Finally, safety is a significant concern, as both oncolytic virotherapy and cancer vaccines can cause side effects, with the former potentially leading to severe or life-threatening reactions in rare instances(239). Recent advancements will change this domain: viruses are now engineered for heightened tumor specificity, some are armed with therapeutic genes to turn tumors into anti-cancer agent producers, and combinations with treatments like immune checkpoint inhibitors are showing synergistic effects. Moreover, refined genetic engineering techniques have improved the safety profiles of these oncolytic viruses, making them more amenable for therapeutic applications. This means that they have reduced virulence in non-target tissues and minimized side effects.” (Page16-17, Line638-662)

Warm Regards,

Guangzhen Wu

Round 2

Reviewer 2 Report

The reviewer thanks the authors for the improvement of the manuscript. The authors have addressed all the questions from the reviewer.